# Endosomal trafficking defects alter neural progenitor proliferation and cause microcephaly

Jacopo A. Carpentieri[1], Amandine Di Cicco[1], Marusa Lampic[1], David Andreau[1], Laurence Del Maestro[2], Fatima El Marjou[1], Laure Coquand[1], Nadia Bahi-Buisson[3,4], Jean-Baptiste Brault[1] & Alexandre D. Baffet [1,5 ✉]

Primary microcephaly and megalencephaly are severe brain malformations defined by reduced and increased brain size, respectively. Whether these two pathologies arise from related alterations at the molecular level is unclear. Microcephaly has been largely associated with centrosomal defects, leading to cell death. Here, we investigate the consequences of *WDR81* loss of function, which causes severe microcephaly in patients. We show that WDR81 regulates endosomal trafficking of EGFR and that loss of function leads to reduced MAP kinase pathway activation. Mouse radial glial progenitor cells knocked-out for *WDR81* exhibit reduced proliferation rate, subsequently leading to reduced brain size. These proliferation defects are rescued in vivo by expressing a megalencephaly-causing mutant form of Cyclin D2. Our results identify the endosomal machinery as an important regulator of proliferation rates and brain growth, demonstrating that microcephaly and megalencephaly can be caused by opposite effects on the proliferation rate of radial glial progenitors.

[1] Institut Curie, PSL Research University, CNRS UMR144, 75005 Paris, France. [2] Epigenetics and Cell Fate (EDC) department, UMR7216, Centre National de la Recherche Scientifique (CNRS), Université de Paris, F-75013 Paris, France. [3] INSERM U1163, Institut Imagine, Necker Hospital, 75015 Paris, France. [4] Pediatric Neurology, Necker Enfants Malades Hospital, Université de Paris, 75015 Paris, France. [5] Institut national de la santé et de la recherche médicale (Inserm), Paris, France. ✉email: alexandre.baffet@curie.fr

Development of the neocortex relies on neural stem cells called radial glial (RG) cells, that generate the majority of cortical neurons[1]. Neuronal production is restricted to a short period during which all excitatory neurons are produced[2]. This leaves little room for compensatory mechanisms to occur, and alterations during this critical period lead to brain malformations[3]. Indeed, the developing neocortex is highly sensitive to perturbations, and a large number of mutations have been described to specifically alter its growth, but not that of other organs[4].

Primary microcephaly is a severe neurodevelopmental disorder characterized by a head circumference that is more than 3 standard deviations (SD) below the mean[5]. The major molecular cause of microcephaly lies in defects in centrosome number[6,7], maturation[8] and mitotic spindle regulation[9–11], leading to apoptotic cell death. In fact, apoptosis appears to be the leading cause of microcephaly in animal models, irrespective of the upstream affected molecular pathway[12–14]. Reduced proliferation rates of progenitors, while proposed to be a putative cause of microcephaly[15], has received much less experimental support. One notable example is the gene encoding IGFR1, which is mutated in syndromic forms of microcephaly, and when deleted in mouse leads to reduced proliferation and small brain size[16,17].

On the opposite end of the spectrum, megalencephaly (MEG) is a neuronal disorder characterized by brain overgrowth (3 SD over the mean)[18]. The causes of megalencephaly are diverse, but activating mutations in the Pi3K-AKT-mTOR and the Ras-MAPK pathways have been identified as important underlying events[19–21]. Mouse and cerebral organoid models for these activated pathways demonstrated increased proliferation of radial glial cells leading to tissue overgrowth[22–24]. Stabilizing mutations in the downstream target Cyclin D2 were also reported, and its ectopic expression in mouse brain stimulated progenitor proliferation[25].

The EGF receptor (EGFR) and its ligands are major regulators of tissue growth[26]. Accordingly, knock-out of EGFR leads to a dramatic atrophy of the cerebral cortex[27]. Progenitor cells appear to become responsive to EGF at mid-neurogenesis, while at earlier stages they rather exhibit FGR2 dependence[28]. Endosomal trafficking of EGFR plays a major role in the regulation of its activity: while most EGFR signaling is believed to occur at the plasma membrane, internalization of EGFR is critical for signal termination[29]. Following endocytosis, internalized cargos follow different trafficking routes including recycling towards the plasma membrane or delivery to lysosomes for degradation[30]. Phosphatidylinositols (PtdIns) are major regulators of this process, defining endosomal compartment identity. Early endosomes are characterized by the presence of the small GTPase RAB5 and PtdIns3P, and late endosomes by RAB7 and PtdIns(3,5)$P_2$[31]. Recently, WDR81 and its partner WDR91 were shown to act as negative regulators of class III phosphatidylinositol 3-kinase (PI3K)-dependent PtdIns3P generation, therefore promoting early to late endosomal conversion[32]. In WDR81 knock-out (KO) HeLa cells, endosomal maturation defects led to delayed EGFR degradation[32].

We recently reported compound heterozygous mutations in the human *WDR81* gene, that result in severe microcephaly at birth that progresses in the first years of life, associated with reduced gyrification of the neocortex[33]. Here, we generated a knock-out mouse model that largely recapitulates the human phenotype. Mutant brains are not only smaller but also display altered neuronal layering. We demonstrate that microcephaly is the result of reduced proliferation rates of radial glial progenitors, but not of cell death. Mechanistically, we show that WDR81 mutation delays EGFR endosomal trafficking and leads to reduced activation of the MAPK signaling pathway. These proliferation defects can be rescued by expressing a megalencephaly-causing mutated cyclin D2, indicating that microcephaly and megalencephaly can be due to opposite effects of the proliferation rates of radial glial cells.

## Results

**WDR81 KO mice display reduced brain size and altered neuronal positioning.** In mice, two WDR81 isoforms have been identified. A long isoform (Isoform 1, 210 kDa) encompassing an N-terminal BEACH domain, a central transmembrane region, and a C-terminal WD40 repeat domain; and a shorter isoform (Isoform 2, 81 kDa) lacking the BEACH domain (Fig. 1a). Measurements of mRNA isolated from embryonic E14.5 cortex extracts indicated that WDR81 isoform 1 was highly dominant, with only trace levels of the short isoform (Fig. 1b). Isoform 1 expression gradually increased from E12.5 to E16.5 (Supplementary Fig. 1a). We generated two WDR81 KO mice using gRNAs targeting the beginning of exon 1 (KO-1, affecting isoform 1), and the end of exon 1 (KO-2, affecting both isoforms) (Fig. 1a). Both lines displayed frameshifts leading to the appearance of a premature STOP codon (Supplementary Fig. 1b). QPCR measurements in KO1 did not reveal any upregulation of isoform 2, indicating an absence of compensation (Fig. 1b). Moreover, a strong reduction of isoform 1 mRNA levels was observed, likely due to non-sense mRNA decay (Fig. 1b). WDR81 homozygote mutant embryos and pups were detected at sub-mendelian rates, and did not live for more than 21 days (Supplementary Fig. 1c).

We then analyzed brain size and organization in WDR81$^{-/-}$ pups. Both KO lines were severely microcephalic, with a reduced hemisphere area at P7 (Fig. 1c, d). Cortical thickness was also greatly reduced (by ~54%), suggesting defects both in tangential and radial expansion of the brain (Fig. 1e, f). As in patients, microcephaly was present at birth and progressed, with reduced cortical thickness and reduced number of NEUN+ neurons at P0 (Fig. 1f, g, h). We next analyzed neuronal positioning in WDR81$^{-/-}$ P7 cortices. The localization of upper layer late-born neurons was severely affected, with a large number of CUX1-positive neurons dispersed throughout the cortex (Fig. 1i). Deeper neurons, which are born earlier during cortical development were however correctly positioned, as indicated by the localization of CTIP2-positive neurons (Fig. 1j). Overall, mice knocked out for WDR81 have reduced brain size and altered neuronal positioning, largely recapitulating the microcephaly and lissencephaly phenotypes reported in humans. These phenotypes were observed for both WDR81 KO lines and we therefore next focused our analysis on KO-1 (referred to as WDR81$^{-/-}$ from here on).

**WDR81 KO alters radial glial progenitor proliferation.** To identify the causes of reduced brain size in WDR81$^{-/-}$ pups, we tested for alterations of neocortex development at embryonic stages. To test for proliferation defects, we first measured the mitotic index of WDR81$^{-/-}$ radial glial progenitors, as defined by the percentage of phospho-Histone H3 (PH3)-positive cells out of total PAX6-positive cells. While at E12.5, proliferation appeared normal, mitotic index of WDR81$^{-/-}$ radial glial progenitors was severely reduced at E14.5 and E16.5 (Fig. 2a). Strikingly, TBR2-positive intermediate progenitors appeared to cycle normally throughout development (Fig. 2b). Therefore, WDR81 mutation specifically alters proliferation of radial glial progenitors at mid and late neurogenic stages. We next analyzed further these proliferation defects and measured the percentage of cells in S phase. A 30-minute BrdU pulse revealed an increased amount of WDR81$^{-/-}$ radial glial progenitors in S-phase (Fig. 2c, d). To test whether this was due to a longer duration of S-phase, we

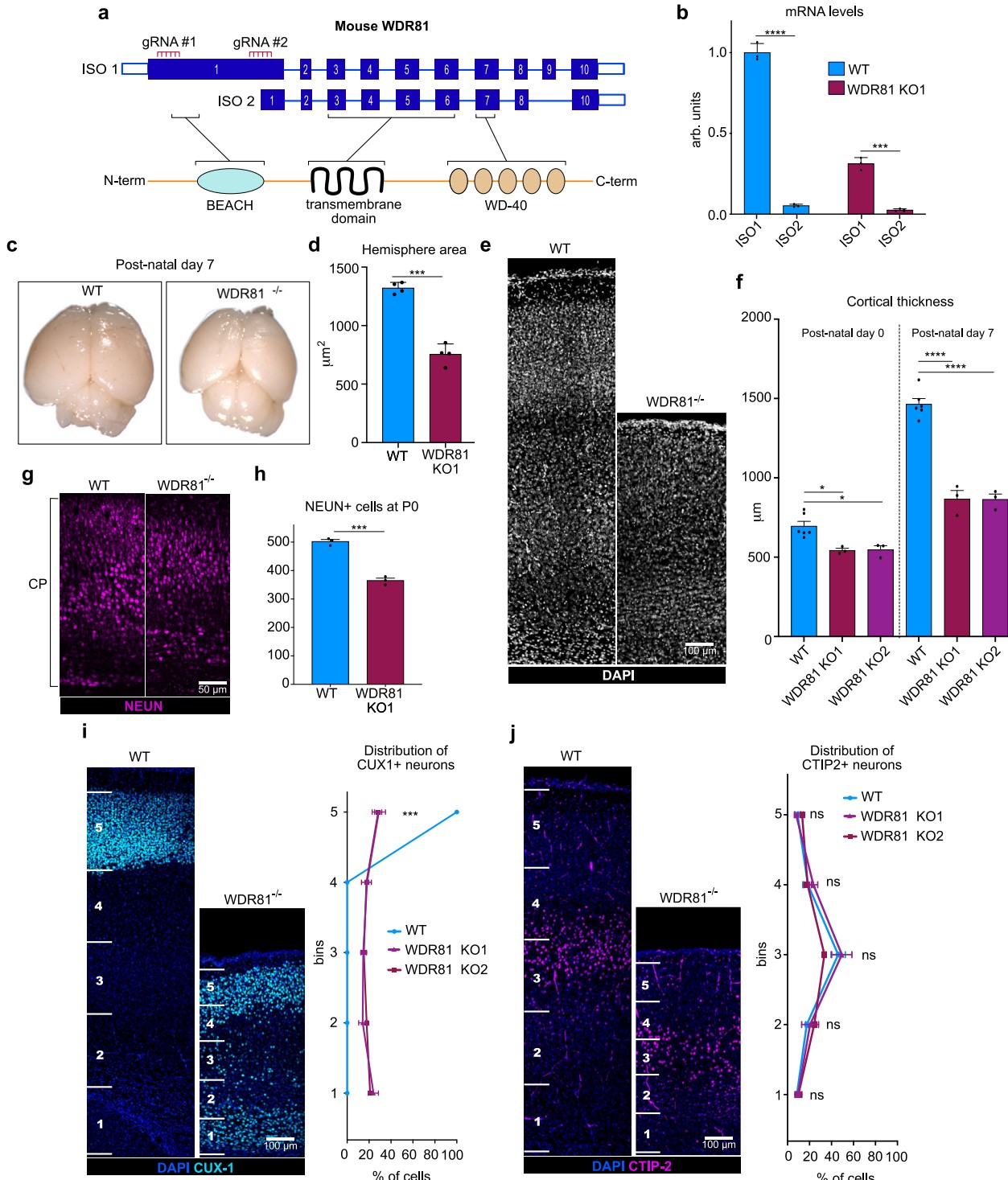

**Fig. 1 WDR81 KO mice display reduced brain size and altered neuronal positioning. a** Schematic representation of mouse WDR81 isoforms and predicted structure. **b** Quantification of WDR81 isoforms 1 and 2 mRNA levels in WT ($p < 0.0001$) and WDR81$^{-/-}$ ($p = 0.0002$) E14.5 cortices ($n = 3$ independent brains for each genotype). Isoform 1 (ISO1) is the dominant isoform and its levels are strongly reduced in WDR81$^{-/-}$ cortices. **c** WDR81$^{-/-}$ postnatal day 7 brains are microcephalic and display reduced cortical surface area as compared to WT brains. **d** Quantification of hemisphere area at P7 in WT and WDR81 KO1 brains ($p < 0.0001$) ($n = 4$ independent brains for each genotype). **e** DAPI staining of P7 WT and WDR81$^{-/-}$ cross sections reveals reduced cortical thickness in mutants. **f** Quantification of cortical thickness in WT, KO1 and KO2 brains at P0 and P7 (At P0, WT vs KO1 $p = 0.0112$; WT vs KO2 $p = 0.0158$. At P7, WT vs KO1 $p < 0.0001$; WT vs KO1 $p < 0.0001$) ($n = 4$ independent brains for each genotype and stage). **g** NeuN staining of WT and WDR81$^{-/-}$ cortical plates (CP) at P0. **h** Quantification of NEUN+ cells in WT and WDR81 KO1 cortical plates at P0 in $600 \times 300\,\mu m$ crops reveals reduced number of neurons at birth ($p = 0.0002$) ($n = 3$ independent brains for each genotype). **i** CUX1 staining in P7 WT and WDR81$^{-/-}$ cortices. Quantification of CUX1+ neuronal positioning reveals dispersion throughout the thickness of the neocortex (Bin 5, $p = 0.0003$) ($n = 5$ independent brains for each genotype). **j** CTIP2 staining in P7 WT and WDR81$^{-/-}$ cortices. Quantification does neuronal positioning defects, with CTIP2+ neurons still concentrated in the third bin ($n = 5$ independent brains for each genotype). **i, j** WT and mutant cortices were divided into 5 bins of equal size to measure neuronal relative positioning, independently of cortical thickness. All data are expressed as mean ± standard deviation (SD). *$p < 0.05$; **$p < 0.01$; ***$p < 0.001$; ****$p < 0.0001$ by two-tailed unpaired $t$ tests.

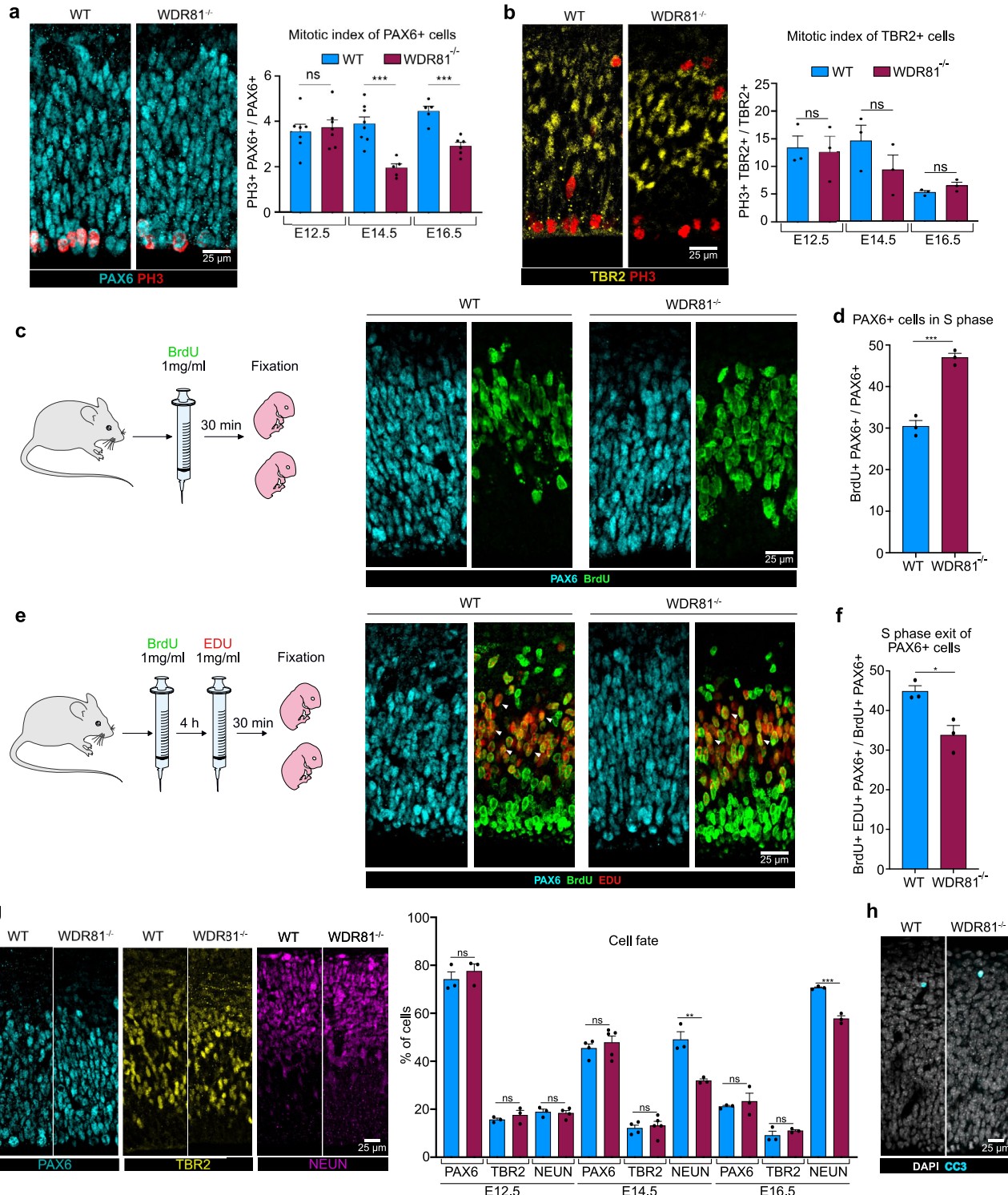

performed a double BrdU-EdU pulse, in order to measure the rate of S-phase exit. Mice were first injected with BrdU, followed by a second injection with EdU 4 h later. This assay revealed a decreased proportion of cells that exited S phase (BrdU+/EdU−) in WDR81$^{-/-}$ brains, indicating a longer S phase in mutant radial glial progenitors (Fig. 2e, f).

An alternative potential cause of reduced brain size is premature differentiation of progenitor cells. To test this, embryonic cortices were stained for PAX6 (radial glial progenitors), TBR2 (intermediate progenitors) and NEUN (neurons) at different developmental stages and the proportion of each cell population was measured. We did

not observe any decrease in the proportion of progenitors out of the total cell population throughout development indicating they did not prematurely differentiate (Fig. 2g). In fact, we even detected a reduction of the proportion of neurons at mid and late neurogenesis (Fig. 2g). Finally, we analyzed apoptotic cell death in WDR81$^{-/-}$ cortices. Staining for cleaved caspase-3 (CC3) did not reveal any increased apoptosis, which remained almost undetectable both in WT and mutant embryos (Fig. 2h). Therefore, reduced brain size in WDR81$^{-/-}$ mice is not the result of premature progenitor differentiation or increased apoptotic cell death, but appears to be a consequence of reduced radial glial progenitor proliferation rates.

**Fig. 2 WDR81 KO alters radial glial progenitor proliferation. a** PAX6 and PH3 double staining in E14.5 WT and WDR81$^{-/-}$ brains. Quantification of the mitotic index of PAX6+ cells reveals decreased proliferation of WDR81$^{-/-}$ radial glial progenitors at E14.5 ($p = 0.0005$) and E16.5 ($p = 0.0003$) ($n = 5$–8 independent brains for each genotype and stage). **b** TBR2 and PH3 double staining in E14.5 WT and WDR81$^{-/-}$ brains. Quantification of the mitotic index of TBR2+ cells indicates that proliferation of WDR81$^{-/-}$ intermediate progenitors in not affected ($n = 3$ independent brains for each genotype and stage). **c** Schematic representation of the BrdU labeling experimental approach and Pax6 and BrdU staining in E14.5 WT and WDR81$^{-/-}$ brains. **d** Quantification of the percentage of BrdU+ PAX6+ out of total PAX6 cells reveals increased number of cells in S phase in WDR81$^{-/-}$ radial glial progenitors at E14.5 ($p = 0.0007$) ($n = 3$ independent brains for each genotype). **e** Schematic representation of the BrdU/EdU double labeling experimental approach and PAX6, BrdU and EdU staining in E14.5 WT and WDR81$^{-/-}$ brains. Arrowheads indicate EdU+ BrdU+ cells. **f** Quantification of the percentage of of BrdU+ EdU− PAX6+ out of the total BrdU+ PAX6+ cells reveals a decreased proportion of cells that exited S phase following BrdU injection in WDR81$^{-/-}$ radial glial progenitors at E14.5 ($p = 0.0174$) ($n = 3$ independent brains for each genotype). **g** Staining for the cell fate markers PAX6 (radial glial progenitors), TBR2 (Intermediate progenitors) NEUN (Neurons) in E14.5 WT and WDR81$^{-/-}$ brains, and quantification of cell fate distribution at E12.5, E14.5 ($p = 0.071$ for NeuN) and E16.5 ($p = 0.0005$ for NeuN). ($n = 3$–5 independent brains for each genotype, staining and stage) **h** Staining for Cleaved Caspase-3 (CC3) and DAPI in E14.5 WT and WDR81$^{-/-}$ brains, showing an absence of apoptosis induction. All data are expressed as mean ± standard deviation (SD). *$p < 0.05$; **$p < 0.01$; ***$p < 0.001$ by two-tailed unpaired $t$ tests.

**Reduced proliferation and EGFR signaling in WDR81 patient cells**. We next tested whether similar proliferation defects could be observed in patient cells mutated for WDR81. Two mutant primary fibroblast lines, derived from skin biopsies, were analyzed and compared to two control fibroblast lines. Both patient lines (Patient1: 1882C-T/3713C-G; Patient2: 1582C-T/Del4036_4041) display compound heterozygous mutations, with one severe mutation (premature STOP) and one point mutation leading to a single amino acid change. The mitotic index of both patient cells was strongly decreased, mimicking the mouse radial glial progenitor phenotype (Fig. 3a, b). As an alternative measurement of proliferation, cells were stained for Ki67, which also revealed a substantial decrease for both patient cell lines (Fig. 3c, d).

In radial glial progenitors, we detected proliferation defects from mid-neurogenesis (Fig. 2a), which fits with the time when these cells start responding to EGF[28]. This observation suggested a potential alteration in the EGFR signaling pathway. In order to test this, we monitored the activity of this signaling pathway in control and patient fibroblasts. Strikingly, we observed that the protein levels of EGFR itself were drastically reduced in both patient cell lines (Fig. 3e, f). We next measured the activation of the mitogen-activated protein kinase (MAPK) signaling pathway in response to EGF stimulation, perfomed after a short (2H) EGF starvation. Consistent with the decreased levels of EGFR, the phosphorylation of ERK was reduced in both patient fibroblasts following EGF pulse (Fig. 3g, h, i and Supplementary Fig. 2a, b). No effect on the levels of phospho-AKT was detected, suggesting no effect on the PI3K pathway (Supplementary Fig. 2c). Therefore, WDR81 patient cells display reduced EGFR levels, leading to a reduced activation of the MAPK signaling pathway upon EGF stimulation.

The levels of EGFR are known to be tightly regulated, through complex feedback loops and the balance between recycling and degradation of the internalized receptor[34]. We therefore asked whether reduced EGFR levels were a consequence of defects within the EGFR pathway itself. To test this, we performed a long-term EGF starvation (24H) in order to abolish EGFR signaling, and subsequently measured the levels of EGFR. In patient cells, starvation rescued EGFR levels to the ones of control cells (Fig. 3j, k). These results indicate that reduced EGFR levels are only seen when the pathway is activated. Given that EGFR internalization upon EGF binding is a major regulator of the pathway, this data points towards intracellular processing defects of EGFR.

**WDR81 is required for endosomal trafficking of EGFR**. WDR81 is known to regulate endosomal maturation as well as autophagic clearance of aggregated proteins (aggrephagy)[32,35].

Importantly, these two functions are independent and act via the WDR81 specific binding partners WDR91 and p62, respectively. We therefore tested whether one of these factors affected neo-cortex development similarly to WDR81. To perform this, we in utero electroporated shRNA-expressing constructs for WDR81, WDR91 and p62 in E13.5 developing brains and analyzed cell distribution at E17.5 (Supplementary Fig. 3a). Consistent with the KO data, WDR81 knock-down (KD) strongly affected neurodevelopment (Fig. 4a, b). In particular, a large fraction of KD cells accumulated in the intermediate zone (IZ), at the expense of the germinal zones and cortical plate. This phenotype was phenocopied by WDR91 KD, but not by p62 KD which did not appear to affect cell distribution (Fig. 4a, b). These results support the endosomal function of WDR81 as a critical player for proper neocortex development.

We next tested whether endosomal defects could be observed in WDR81 KO radial glial progenitors. Staining for various endolysosomal compartments revealed a specific alteration of EEA1+ early endosomes, which appeared strongly enlarged (Fig. 4c). Quantification of their size confirmed this observation, revealing a 63% average increase (Fig. 4e). To test whether this is a conserved feature of WDR81 patient cells, we measured early endosome size in mutant fibroblasts. Again, EEA1+ endosomes were found to be swollen, with an increased proportion of large endosomes (>0.5 μm) (Fig. 4d, f). These results are consistent with previous observations made in KO HeLa cells and demonstrate a role for WDR81 in negative regulation of Class III Pi3K[32].

Because these endosomal defects are a potential cause of altered EGFR signaling, we next tested whether EGFR endosomal trafficking was affected in WDR81 mutant cells. Cells were first starved for 24 h to restore EGFR to the levels of control cells (See Fig. 3j), and subsequently pulsed with fluorescent EGF$^{555}$, to monitor internalization and clearance of EGF-bound EGFR. In both patient cells, EGF$^{555}$ was shown to accumulate longer within EEA1+ early endosomes (Fig. 4g). Quantification of the colocalization between EGF$^{555}$ and EEA1 revealed that this delay was particularly important 120 minutes after EGF internalization (Fig. 4h). We confirmed these defects in vivo, where EGFR was observed to accumulate in larger and more numerous intracellular foci within KO radial glial progenitors (Supplementary Fig. 3b, c, d). Finally, we tested if endosomal maturation defects also led to impaired recycling, using a Transferrin uptake assay. Transferrin$^{546}$ was observed to be strongly delayed into EEA1+ early endosomes of patient cells after 1 h, indicating defective processing of the Transferrin receptor (Supplementary Fig. 3e, f). Overall, these results show that WDR81 is critical for endosomal homeostasis and trafficking of internalized EGFR following EGF binding.

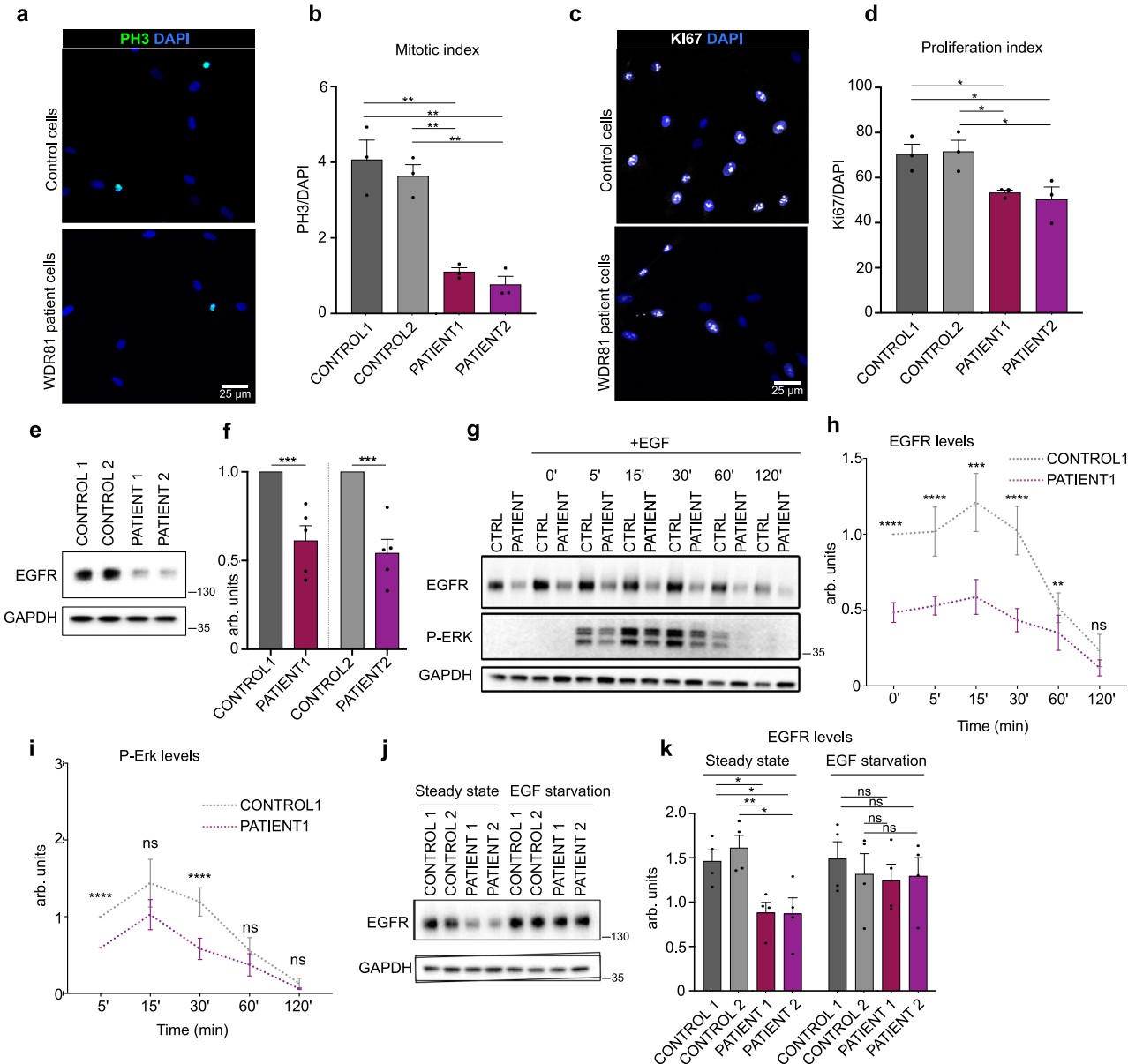

**Fig. 3 Reduced proliferation and EFGR signaling in WDR81 patient cells. a** PH3 and DAPI staining in control and WDR81 patient fibroblasts.
**b** Quantification of the percentage of PH3+ cells reveals decreased mitotic index in patient cells (control1-patient1 $p = 0.0051$; control1-patient2
$p = 0.0043$; control2-patient1 $p = 0.0015$; control2-patient2 $p = 0.0016$) ($n = 3$ independent experiments). **c** Ki67 and DAPI staining in control and
WDR81 patient fibroblasts. **d** Quantification of the percentage of Ki67+ cells shows decreased proliferation in patient cells (control1-patient1 $p = 0.0207$;
control1-patient2 $p = 0.0476$; control2-patient1 $p = 0.0245$; control2-patient2 $p = 0.0473$) ($n = 3$ independent experiments). **e** Western Blot for EGFR in
control and WDR81 patient fibroblasts ($n = 3$ independent experiments). **f** Quantification reveals a strong reduction of EGFR levels in patient cells
(control1-patient1 $p = 0.0017$; control1-patient2 $p = 0.0004$) ($n = 5$ independent experiments). **g** Time course of EGFR and P-ERK levels in control1 and
WDR81 patient1 fibroblasts following an EGF pulse. **h** Quantification of EGFR levels, normalized to control levels at T0 in control1 and patient1 (T0
$p = p < 0.0001$; T5 $p = p < 0.0001$; T15 $p = 0.0005$; T30 $p < 0.0001$; T60 $p = 0.0075$) ($n = 5$ independent experiments). **i** Quantification of P-ERK levels,
normalized to control levels at T5 in control1 and patient1 (T5 $p < 0.0001$; T30 $p < 0.0001$) ($n = 5$ independent experiments). **j** Western Blot for EGFR in
control and WDR81 patient fibroblasts at steady state (+EGF) and cultivated for 24H in the absence of EGF. **k** Quantification reveals a restoration of EGFR
levels following starvation (at steady state, control1-patient1 $p = 0.0161$; control1-patient2 $p = 0.0368$; control2-patient1 $p = 0.0075$; control2-patient2
$p = 0.0176$) ($n = 4$ independent experiments). All data are expressed as mean ± standard deviation (SD). *$p < 0.05$; **$p < 0.01$; ***$p < 0.001$; ****$p < 0.0001$
by two-tailed unpaired $t$ tests.

**Megalencephaly-causing mutation rescues progenitor proliferation in WDR81 mutant brains**. Our results indicate that trafficking defects of EGFR can arise from mutations in WDR81, and lead to reduced activation of the MAPK signaling pathway. They further show that reduced radial glial progenitor proliferation is a cause of primary microcephaly. Megalencephaly is characterized by brain overgrowth and can be due to increased cell proliferation

during development[18]. Major causes include gain-of-function mutations in AKT3 and its downstream target Cyclin D2[25,36]. Together, these data suggest that microcephaly and megalencephaly can be the consequence of opposite effects on the proliferation rates of radial glial progenitors. To further test this, we analyzed the effect of a megalencephaly-causing Cyclin D2[Thr280Ala] mutant on the proliferation of WDR81 KO radial glial progenitors. Degradation-

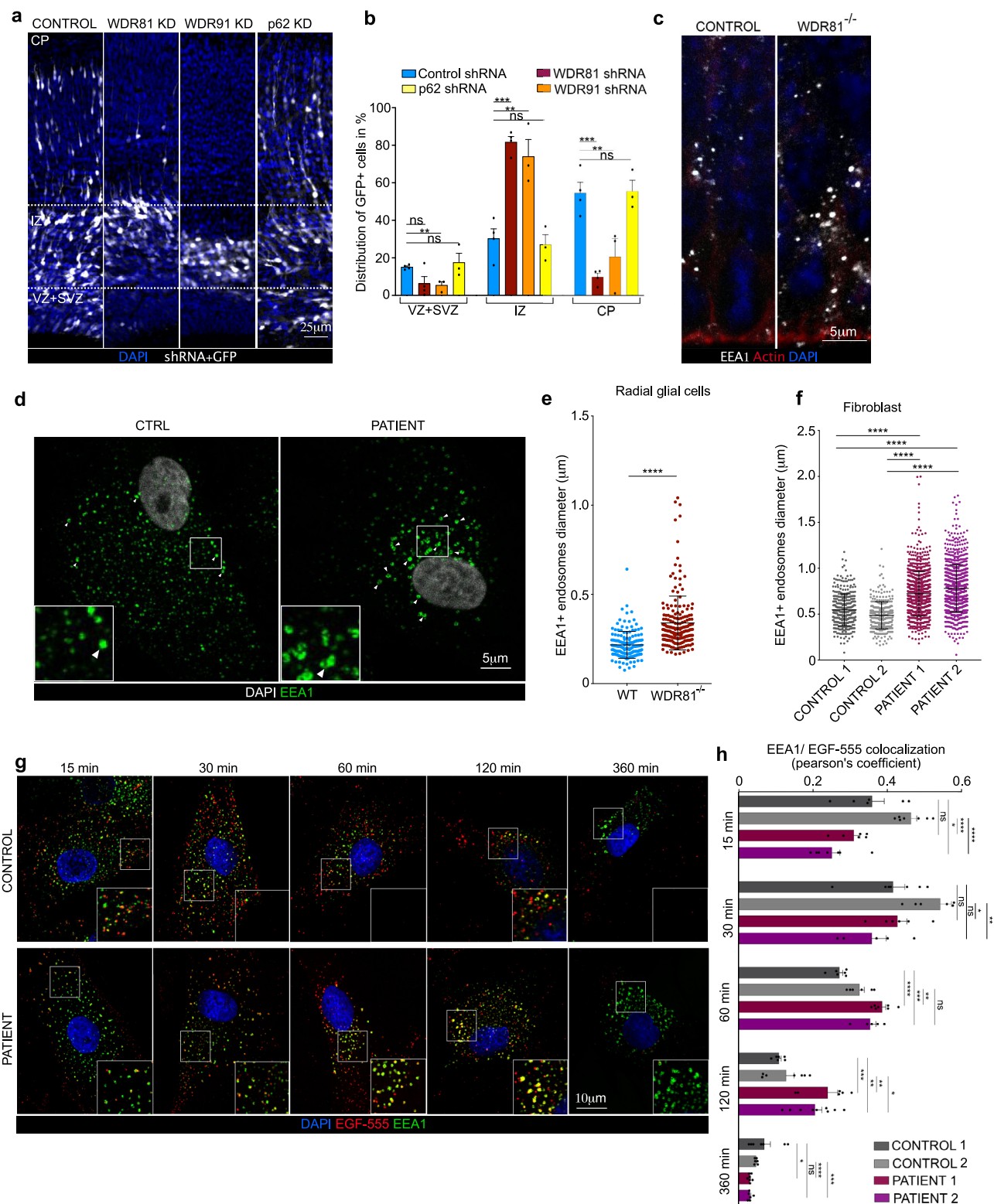

resistant Cyclin D2^Thr280Ala, WT Cyclin D2 or a control vector were expressed using in utero electroporation in WT and WDR81-mutant mice brains at E14.5, and the mitotic index of PAX6 cells was measured at E16.5. Following expression of the control empty vector, we confirmed the reduced mitotic index in WDR81$^{-/-}$ brains (Fig. 5a, b). Moreover, expression of Cyclin D2^Thr280Ala in WT brain increased mitotic index, indicating that this megalencephaly-causing mutation indeed stimulates radial glial

progenitor proliferation rates (Fig. 5a, b). Strikingly, expression of degradation-resistant Cyclin D2^Thr280Ala in WDR81$^{-/-}$ brains rescued the mitotic index reduction (Fig. 5a, b). WT cyclin D2 was also able to rescue proliferation, although to a lesser extent, indicating that large amounts of this protein, either due to overexpression or impaired degradation is able to restore proliferation (Fig. 5a, b). Together, these results indicate that a megalencephaly-causing mutation can overcome the effect of a microcephaly-causing

**Fig. 4 WDR81 is required for endosomal trafficking of EGFR. a** Expression of control shRNA and shRNA-mediated knockdown constructs for WDR81, WDR91 and p62, together with pCAG-GFP. Plasmids were delivered by in utero electroporation at E13.5 and analysis was performed at E16.5. Ventricular Zone and Sub-Ventricular Zone (VZ + SVZ), Intermediate Zone (IZ) and Cortical Plate (CP) were identified based on DAPI staining. **b** Quantification of electroporated cell distribution reveals major accumulation in the IZ following WDR81 and WDR91 knockdown (VZ + SVZ, WDR91 shRNA $p = 0.0025$; IZ, WDR81 shRNA $p = 0.0001$, WDR91 shRNA $p = 0.067$; CP, WDR81 shRNA $p = 0.0003$, WDR91 shRNA $p = 0.0234$) ($n = 3–4$ independent brains per condition). **c** Ventricular zone of E14.5 WT and WDR81$^{-/-}$ mice cortices stained for EEA1 and Actin. **d** Control and WDR81 patient fibroblasts stained for EEA1. **e** Quantification of individual EEA1+ early endosomes in WT and WDR81$^{-/-}$ VZ in 200 × 100 µm crops reveals increased size in mutant brains ($p < 0.0001$) ($n = 3$ independent experiments). **f** Quantification of individual EEA1+ early endosomes in control and WDR81 patient fibroblasts reveals increased size in mutant cells (For all control-patient couples $p < 0.0001$) ($n = 3$ independent experiments). **g** EGF$^{555}$ uptake assay in control and WDR81 patient fibroblasts, and stained for EEA1. **h** Quantification of EGF$^{555}$ and EEA1 colocalization during EGF$^{555}$ uptake reveals prolonged colocalization between EGF and early endosomes in WDR81 patient cells (At T15, C1 vs P2 $p = 0.0158$; C2 vs P1 $p < 0.0001$; C2 vs P2 $p < 0.0001$. At T30, C2 vs P1 $p = 0.0149$, C2 vs P2 $p = 0.0033$. At T60, C1 vs P1 $p < 0.0001$; C1 vs P2 $p = 0.001$; C2 vs P1 $p = 0.0028$. At T120, C1 vs P1 $p = 0.0008$; C1 vs P2 $p = 0.0011$; C2 vs P1 $p = 0.0072$; C2 vs P2 $p = 0.0148$. At T360, C1 vs P1 $p = 0.0349$; C2 vs P1 $p < 0.0001$. C2 vs P2 $p = 0.0001$) ($n = 7$ independent experiments). All experiments were performed at least three independent times. All data are expressed as mean ± standard deviation (SD). *$p < 0.05$; **$p < 0.01$; ***$p < 0.001$; ****$p < 0.0001$ by two-tailed unpaired $t$ test (A & H) and Mann–Whitney tests (E & F).

mutation on the proliferation of radial glial progenitors. These two pathologies can therefore arise from a highly related cause: an imbalance in cell cycle regulation leading either to reduced brain growth or to brain overgrowth (Fig. 5c).

## Discussion

In this study, we investigated the mechanisms by which mutation in the *WDR81* gene leads to severe microcephaly in patients. We show that KO mouse recapitulates many features of the phenotype previously observed in patients and that the endosomal maturation function of WDR81 is critical for neocortex development. WDR81 is required for endosomal clearance of internalized EGFR and normal activation of the mitogenic MAPK signaling pathway. In the absence of WDR81, the proliferation rate of radial glial cells is affected, leading to reduced brain size. Importantly, cell death does not appear to contribute to this phenotype. Proliferation defects can be rescued by the expression of a megalencephaly-causing mutated cyclin D2, highlighting a tight functional link between these two pathologies.

Membrane trafficking has been poorly investigated in radial glial cells, albeit its predicted implication in many important processes including cargo polarized transport, secretion of extracellular matrix components, or endocytic processing of surface receptors for lysosomal degradation or recycling. We show here that the endosomal maturation machinery plays a critical role in the processing of internalized EGFR in RG cells, and is required for their proliferation. Neurogenesis depends on EGFR activity, with radial glial cells becoming responsive to EGF from mid-neurogenesis[27,28]. We find that WDR81 expression rises at from E14.5 and that KO RG cells are specifically affected at E14.5 and E16.5 stages of development. Why the proliferation rate of IPs was not affected is unclear but EGF is secreted into the cerebrospinal fluid from the choroid plexus and apical contact may be critical for responsiveness[37,38]. EGFR was previously reported to be asymmetrically inherited during radial glial cell division, generating a daughter cell with higher proliferative potential[37]. Later in development, EGFR also acts as an important regulator of astrocyte differentiation[39]. Our data point to the intracellular processing of EGFR as an important level of control for the regulation of proliferation in radial glial cells. WDR81 is likely to affect the trafficking of other cargos[40], which may also impact radial glial cell proliferation. Moreover, the trafficking of neuronal cargos, such as adhesion molecules, is likely to lead to the altered neuronal positioning observed in KO mice, and to the lissencephaly phenotype in human.

In principle, microcephaly can be the consequence of premature progenitor differentiation, reduced proliferation rates, or cell death. While centrosomal defects leading to apoptosis have been

described, reduced proliferations rates have received little experimental support[16]. In mouse, RG cells produce eight to nine neurons during a short neurogenic period, before differentiating[2]. We show here that *WDR81* mutation does not affect the modes of division of RG cells nor cell survival, but act solely through a reduction of their proliferation rate, leading to reduced brain size. This highlights the absence of compensatory mechanisms in the developing neocortex, where all neurons must be produced in a short temporal window. During corticogenesis, G1 lengthening is associated with increased neurogenic divisions at the expense of symmetric amplifying divisions[41,42]. We did not detect such cell fate changes in WDR81$^{-/-}$ brains. This is likely due to the fact that the proliferation rate of mutant RG cells is only affected at a stage where the vast majority of cells already perform neurogenic divisions[1]. At the macroscopic level, microcephaly and megalencephaly appear as opposite phenotypes. We show here that they can be due to opposite effects on the proliferation rates of RG cells, and can therefore be viewed as two sides of the same coin.

## Methods

**Animals**. All experiments involving mice were carried out according to the recommendations of the European Community (2010/63/UE). The animals were bred and cared for in the Specific Pathogen Free (SPF) Animal Facility of Institut Curie (agreement C 75-05-18). All animal procedures were approved by the ethics committee of the Institut Curie CEEA-IC #118 and by French Ministry of Research (2016-002). Animals were housed at a temperature of 22 °C, 50% humidity and a 12/12 hour light/dark cycle.

**Guide RNA selection and preparation**. gRNA sequences targeting exon 1 of WDR81 have been identified and selected using the online software CRIPSOR (crispor.tefor.net). Forward and reverse oligonucleotides were annealed and cloned into px330 plasmid. To generate Cas9 mRNA and gRNA, in vitro transcriptions were performed on the Cas9 pCR2.1-XL plasmid and gRNA plasmids, using the mMESSAGE mMACHINE T7 ULTRA kit and the MEGAshortscript T7 kit (Life Technologies), respectively. Cas9 mRNA and sgRNAs were then purified using the MEGAclear Kit (Thermo Fisher Scientific) and eluted in RNAse-free water. The gRNA and Cas9mRNA quality were evaluated on agarose gel.

**Generation of WDR81 knock-out mice**. Eight-week-old B6D2F1 (C57BL/6 J × DBA2) females from Charles River France, were superovulated by intraperitoneal (i.p.) administration of 5 IU of Pregnant Mare Serum Gonadotropin followed by an additional i.p. injection of 5 IU Human Chorion Gonadotropin 48 h later. Females were mate to a stud male of the same genetic background. Cytoplasmic microinjection was performed into mouse fertilized oocytes using Cas9 mRNA and sgRNA at 100 ng/µl and 50 ng/µl, respectively in Brinster buffer (10 mM Tris-HCl pH 7.5; 0.25 mM EDTA). Microinjected zygotes were cultured in Cleave medium (Cook, K-RVCL-50) at 37 °C under 5% CO2 and then implanted at one cell stage into infundibulum of E0.5 NMRI pseudo-pregnant females (25–30 injected zygotes per female). According to the genotyping strategy, 3 mice showed modified allele out of a total of 22 pups. The founders were then backcrossed to C57BL6/J.

**Genotyping WDR81$^{-/-}$ animals**. Mice DNA was extracted from a piece of ear (adult) or tail (dissected embryos), put at 96 degrees in lysis tampon overnight. The DNA was then amplified via PCR using WDR81 specific primers: for KO1

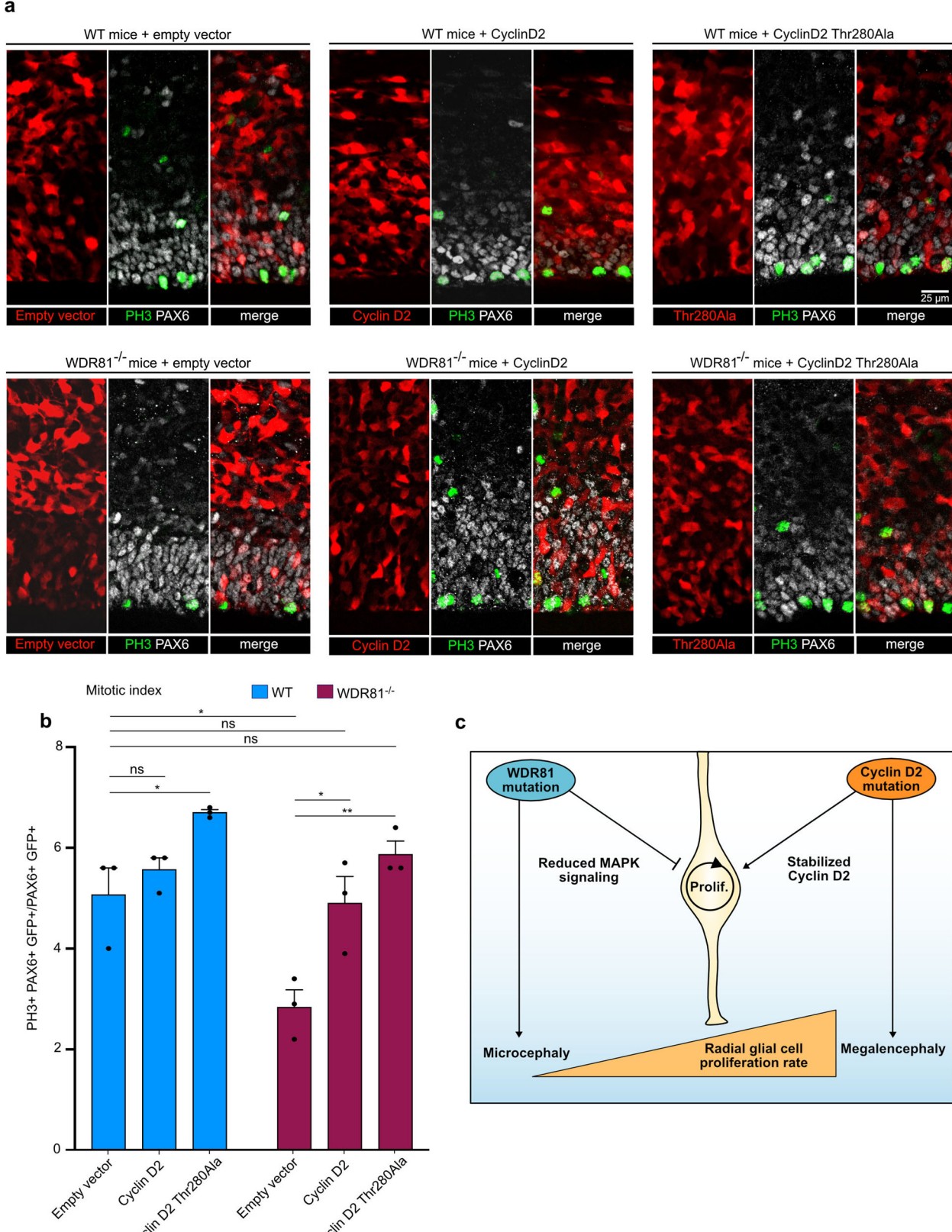

Forward: GGCGGAAAGTGGTTCTTACA, Reverse: AGCCACCTCCTGCATG AACC; for KO2 Forward: GGCTTGTAGTGGTTCTGTAC, Reverse: GATCC TTCTGCATTCCAA. For KO1 the amplicon was purified using the nucleospin purification kit (Machenery and Nagel) and then exposed to the restriction enzyme AfeI (New England Bioscience). The restriction enzyme only cuts the mutant DNA giving rise to two DNA pieces of 200 bp. For KO2, the amplicon was sanger-sequenced using the GATC-Eurofins platform.

**Real-time reverse-transcription PCR**. Wild type and WDR81$^{-/-}$ cortices were dissected at E14.5 in 1 ml of TRIZOL (Thermo Fisher 15596026). The mRNA was isolated as follows: TRIZOL+ sample solution was exposed to chlorophorm for 7 min at room temperature and centrifuged at 15.000 g for 30 min at 4 degrees. The translucent solution formed was then transferred in 1 ml of isopropanol, incubated 7 min at room temperature and centrifuged at 4 degrees 10.000 g. The pellet of nucleic acid formed was then washed in ethanol 70% and centrifuged 5 min at

**Fig. 5 Undegradable Cyclin D2$^{Thr280Ala}$ rescues WDR81$^{-/-}$ proliferation index. a** Expression of Cyclin D2, Cyclin D2$^{Thr280Ala}$ and empty vector in WT and WDR81$^{-/-}$ brains. Constructs were in utero electroporated at E14.5, and brains were fixed at 16.5 and stained for PAX6 and PH3. **b** Quantification of the percentage of mitotic (PH3+) electroporated radial glial cells (PAX6+) out of total electroporated radial glial cells reveals rescue of mitotic index in WDR81$^{-/-}$ cells expressing Cyclin D2$^{Thr280Ala}$ (WT-empty vector vs WT-Cyclin D2$^{Thr280Ala}$ $p = 0.0382$; WT-empty vector vs WDR81$^{-/-}$ empty vector $p = 0.0247$; WDR81$^{-/-}$ empty vector vs WDR81$^{-/-}$ Cyclin D2 $p = 0.031$; WDR81$^{-/-}$ empty vector vs WDR81$^{-/-}$ Cyclin D2$^{Thr280Ala}$ $p = 0.0023$) ($n = 3$ independent brains per genotype and condition). **c** Model. WDR81 loss of function leads to reduced activation of the MAPK signaling pathway downstream of EGFR, to reduced radial glial progenitor proliferation, and to microcephaly. Gain of function in the Pi3K-AKT pathway or stabilizing mutations in Cyclin D2 lead to increased radial glial progenitor proliferation, and to megalencephaly. Cyclin D2 mutants can rescue proliferation defects in WDR81$^{-/-}$ brains, indicating that these two pathologies can arise from opposite effects on the proliferation rates of radial glial progenitor. All data are expressed as mean ± standard deviation (SD). *$p < 0.05$; **$p < 0.01$ by two-tailed unpaired $t$ tests.

10.000 g at 4 degrees. The pellet was then resuspended in water. The nucleic acids solution was purified from DNA using TURBO DNA-free Kit (Thermofisher). The mRNA obtained was then retrotranscribed using the RT reverse transcription Kit (Thermofisher). Real time RT-PCR was performed using the qPCR Master Mix kit (Thermofisher) and the WDR81, WDR91 and p62 Forward/Reverse primers; GAPDH gene was used for internal control and for normalization. Primers used for WDR81 isoform 1 are forward: AGTGGATCCTTCAGACAGCC, Reverse: GAAG CCAGCCACAACACTC. Primers used for WDR81 isoform 2 are Forward: AGTG GATCCTTCAGACAGCC, Reverse: CTGACTTGTAGTGGTGCGTG. Primers used for WDR91 are forward: AGTGCTGAGCCAAGAAGAGT, Reverse: CTAG GGAGAGCAGTGGTGAC. Primers used for p62 are forward: TATCTTCTGGG CAAGGAGGA, Reverse: TGTCAGCTCCTCATCACTGG.

**In utero electroporation of mouse embryonic cortex**. Pregnant mice at embryonic day 13.5 or 14.5 were anesthetized with isoflurane gas, and injected subcutaneously first with buprenorphine (0.075 mg/kg) and a local analgesic, bupivacaine (2 mg/kg), at the site of the incision. Lacrinorm gel was applied to the eyes to prevent dryness/irritation during the surgery. The abdomen was shaved and disinfected with ethanol and antibiotic swabs, then opened, and the uterine horns exposed. Plasmid DNA mixtures were used at a final concentration of 1 µg/µl per plasmid, dyed with Fast Green and injected into the left lateral ventricle of several embryos. The embryos were then electroporated through the uterine walls with a NEPA21 Electroporator (Nepagene) and a platinum plated electrode (5 pulses of 50 V for 50 ms at 1 second intervals). The uterus was replaced and the abdomen sutured. The mother was allowed to recover from surgery and supplied with painkillers in drinking water post-surgery. Electroporated brains were harvested at E16.5 and E17.5.

**Immunostaining of brain slices**. Mouse embryonic brains were dissected out of the skull, fixed in 4% Pfa for 2 h, and 80 µm-thick slices were prepared with a Leica VT1200S vibratome in PBS. Slices were boiled in citrate sodium buffer (10 mM, pH6) for 20 min and cooled down at room temperature (antigen retrieval). Slices were then blocked in PBS-Triton X100 0.3%-Donkey serum 2% at room temperature for 2 h, incubated with primary antibody overnight at 4 °C in blocking solution, washed in PBS-Tween 0.05%, and incubated with secondary antibody overnight at 4 °C in blocking solution before final wash and mounting in aqua-polymount. Imaging was performed on a fully motorized spinning disk wide microscope driven by Metamorph software (Molecular Devices) and equipped with a Yokogawa CSU-W1 scanner unit to increase the field of view and improve the resolution deep in the sample. Image analysis, modifications of brightness and contrast were carried out with Fiji. Statistical analysis was carried out with Prism. Figures were assembled in Affinity Designer.

**Brdu/Edu labeling**. For BrDU labeling experiments, BrDU (Invitrogen B23151) was injected at 50 mg/kg intraperitoneally 30 min prior to harvesting embryos. For BrDU/EDU labeling experiments, BrDU was injected at 50 mg/kg intraperitoneally 4 h prior to harvesting embryos, and EdU (Thermofisher Click-iT EdU Alexa Fluor 555) was injected at 50 mg/kg 30 min prior to harvesting embryos. After fixation, brain slices were incubated in 2 N HCL for 30 min at 37 degrees and then washed 3 times with PBS prior to immunostaining.

**WDR81 patient cells and immunostaining**. Control and WDR81 mutant primary fibroblasts were provided by Institut Imagine, Paris. The genotype of patient 1 cells was compound heterozygote 1882C > T/3713 C > G and the genotype of patient 2 cells was compound heterozygote 1582 C > T/4036_4041dup. Cells were grown in OPTIMEM + 10%FBS at 37 degrees in humid air containing 5% $CO_2$. Fibroblasts were fixed in 4% paraformaldehyde for 20 min, treated with 50 mM NH4Cl for 10 min, washed three times with PBS and left in a blocking solution (PBS 1% donkey serum 0.1% Triton X) for 30 min. Cells were then incubated 1 h at room temperature with primary antibodies, washed three times in PBS and incubated for

45 min at room temperature in blocking solution with Alexa Fluor coupled secondary antibodies. Cells were then washed and mounted.

**Antibodies**. Primary antibodies used: mouse anti Ctip2 (Abcam ab18465, 1/300), rabbit anti Pax6 (Biolegend 901301 1/500), Sheep anti TBR2/EOMES (R&D system AF6166 1/500), rabbit anti NEUN (Abcam ab177487 1/1000), goat anti Phospho Histone3 (Santa Cruz SC-12927 1/1000), rabbit anti CUX1 (Santa Cruz, discontinued 1/100), rabbit anti BRDU (Abcam AB152095 1/500), rabbit cleaved caspase-3 (CST 9661, 1/2000), rabbit anti Ki67 (abcam ab15580, 1/500), rabbit anti EGFR (CST 4267, 1/100), mouse anti p-ERK (CST 9106, 1/200), rabbit anti GAPDH (Sigma–AldrichG9545, 1/5000), anti p-AKT (CST 4060, 1/200) and mouse anti EEA-1 (BD biosciences 610457, 1/500). Secondary antibodies used: donkey Alexa Fluor 488 anti-mouse (1/250), anti-rabbit, anti-goat (Jackson laboratories 715-545-150, 711-165-152, 715-605-152), donkey Alexa Fluor 555 (1/250) anti-mouse, anti-rabbit, anti-goat (Jackson laboratories 715-545-150, 711-165-152, 715-605-152), donkey Alexa Fluor 647 (1/250) anti-mouse, anti-rabbit, anti-goat (Jackson laboratories 715-545-150, 711-165-152, 715-605-152).

**Expression constructs and shRNAs**. For WDR81 knockdown experiments, WDR81 shRNA was provided by Genecopoeia$^{TM}$. The small interfering RNA sequence was ggagataagcaattggacttc and was cloned in psi-mU6.1 vector coexpressing mcherryFP. For WDR91 knockdown experiments shRNA was provided by Tebu-bio (217MSH024100-mU6). For p62 knockdown experiments shRNA was provided by Tebu-bio (217CS-MSH079315-mU6-01). Constructs were co-injected with GFP-pCagIG (Addgene 11159) at a concentration of 1 ug/ul. Plasmids were introduced in the in vivo developing cortex by intraventricular injection and electroporation. For validation of shRNA efficiency, Neuro2A cells (ATCC, CCL-131) were transfected with WDR81, WDR91, p62 and scramble control plasmids using Lipofectamin-3000 (Thermo Fisher Scientific) and lyzed in TRIZOL after 3 days for QPCR. For WDR81 rescue experiments, Cyclin D2 and Cyclin D2 Thr280Ala were synthetized in vitro (Genescript). They were then cloned into GFP-pCagIG (Addgene 11159) after digestion by restriction enzymes EcoRI and EcoRV.

**EGF pulse assay, EGF$^{555}$ uptake assay and Transferrin$^{546}$ uptake assay**. For EGF pulse assay, fibroblast cultures were EGF starved for two hours before the assay. EGF was added directly to the culture medium at 0.1 mg/ml. Cells were then harvested at 0, 5, 15, 30, 60, 120 min and processed for protein extraction. Proteins were then mixed with 4x Leammli (Biorad) and BME solution and used for Western Blot analysis. For EGF$^{555}$ pulse assay, fibroblast cells were cultivated on glass coverslips; cells were starved for 24 h and then exposed to 0,1 mg/ml EGF$^{555}$ (Thermo Fisher). Cells were fixed in paraformaldehyde 4% at 15, 30, 60, 120, 360 min and used for immunostaining. For Transferrin$^{546}$ uptake assay, fibroblast cells were cultivated on glass coverslips; cells were serum-starved for 1 h and then exposed to at 0.1 mg/ml Transferrin$^{546}$ (Thermo Fisher). Cells were fixed in paraformaldehyde 4% at 15, 30, 60, 120, 360 min and used for immunostaining.

**Statistical analysis**. Quantitative data are described as mean ± standard deviation (SD) for $n \geq 3$. No data were excluded from the analyses and the experiments were not randomized. Statistical analysis was performed using two-tailed unpaired Student's $t$ test using GraphPad Prism 9 software (GraphPad Software, San Diego, CA, USA). P values lower than 0.05 were considered statistically significant.

**Reporting summary**. Further information on research design is available in the Nature Research Reporting Summary linked to this article.

## Data availability
The original pictures of WB membranes are provided in the file "Supplementary information_WB membranes.pdf" and all quantifications are available in the "Quantifications_Carpentieri et al.xlsx" file, within the Source Data File. The

immunofluorescence data that support the findings of this study, due to their large size, are available from the corresponding author (alexandre.baffet@curie.fr) or from the first author (jacopocarpentieri@gmail.com) upon request. WDR81 KO mice strains will be shared upon request. Source data are provided with this paper.

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

## Acknowledgements

We acknowledge Institut Curie, member of the French National Research Infrastructure France-BioImaging (ANR10-INBS-04) and the Nikon BioImaging Center (Institut Curie, France). We thank Renata Basto, Veronique Marthiens, Iva Simeonova, Cedric Delevoye (I. Curie) and Fiona Francis (IFM), for helpful discussions and critical reading of the manuscript. J.A.C. was funded by the IC3i Institut Curie doctoral program founded by the European Union's Horizon 2020 research and innovation program under the Marie Sklodowska-Curie actions grant agreement and from Fondation de la Recherche Medicale (FRM). A.D.B. is an Inserm researcher. This work was supported by the CNRS, Institut Curie, the Ville de Paris "Emergences" program, Labex CelTisPhyBio (11-LBX-0038) and PSL university.

## Author contributions

J.A.C. and A.D.B. conceived the project. J.A.C. & A.D.B. analyzed the data. J.A.C., J.B.B. & A.D.B. wrote the manuscript. J.A.C., A.D.C. and M.L. did most of the experimental procedures. J.A.C., L.D.M. and F.E.M. created the mutant mouse lines. D.A. supervised the mouse mutant colonies and performed all the crossings. L.C. and J.A.C. set up in utero electroporation experiments. N.B.B. provided biological samples of the affected patients A.D.B. supervised the project.

## Competing interests

The authors declare no competing interests.
