## [Peer Review File · Nature Communications]

Endosomal trafficking defects alter neural progenitor proliferation and cause microcephalyREVIEWER COMMENTS

Reviewer #1 (Remarks to the Author):

In this study, the authors provided data to suggest that Wdr81-mediated endosomal trafficking of EGFR plays a role in radial glial progenitor cell proliferation, and that Wdr81 deficiency leads to microcephaly. Although these findings provided additional evidence that WDR81 is important for neurogenesis, previous work has already shown that WDR81 is required for endosomal trafficking of EGFR and loss of WDR81 impaired neurogenesis in postnatal brain in mice. Thus, the current manuscript reports that WDR81 serves a somewhat similar purpose in radial glial cells as it does in HeLa cells and adult neural stem cells. Overall, the conceptual and mechanistic advance made by this study is limited.

1. Microcephaly is a condition defined as a small head circumference present at birth (congenital or primary microcephaly) or later in life (postnatal or secondary microcephaly). Congenital microcephaly is usually caused by a decrease in the number of neurons generated during neurogenesis; and postnatal microcephaly is thought to result from the postnatal reduction of dendritic processes, myelination, and synaptic connections. In Wdr81 mutant mice, the numbers of Pax6 apical progenitor and Tbr2 basal progenitor cells and subsequent neurogenesis are quite normal. While the author only showed that the Wdr81 mutant mice displayed reduced brain size at P7, the etiology of microcephaly in Wdr81 mutant need further analysis.

2. Wdr81 mutation specifically alters proliferation of radial glial progenitors at mid and late, but not early, neurogenic stages. Is this a result of the distinct tempo-spatial expression of WDR81 during embryonic brain development?

3. Cyclin D2 being localized to the endfoot of APs acts as a “basal fate determinant.” During asymmetric cell division, one of the daughter cells inherits its basal process, which automatically leads to asymmetrical inheritance of the cyclin D2 protein between the daughter cells. The daughter cell with cyclin D2 then becomes an AP, and the other without cyclin D2 becomes a neuronal cell or an intermediate progenitor (Tsunekawa et al 2012 EMBO J). In this study, the author showed that the Cyclin D2 overexpression rescued the progenitor proliferation in Wdr81 mutant mice. However, is Cyclin D2 expression affected in Wdr81 mutant radial glial cells? In addition, is the asymmetric division of radial glial cells affected in Wdr81 mutant mice?

Reviewer #2 (Remarks to the Author):

Carpentieri et al present data investigating the role of WDR81 during microcephaly. They generate mice lacking WDR81 and investigate brain size and neuronal cell development, growth and survival. They

show that mice lacking WDR81 exhibit a microcephaly phenotype and they attribute this effect to a prolonged S-phase of radial glial progenitors. The authors also investigate fibroblasts derived from patients and harbouring WDR81 mutations. They conclude from these experiments that in patient cells, EGFR levels, signalling and trafficking are altered when compared to control cells. Finally, the authors show that the expression of a degradation-resistant cyclinD mutant can reverse the mitotic defect observed in WDR81^{-/-} mice.

Overall, this is an interesting study. However, some of the findings are not investigated in depth and require further examination. More specific comments are as follows:

The authors show WDR81 RNA levels but have not confirmed the efficiency of WDR81 knockout as some cells may still express full length protein. If antibodies are not available to do so, can the authors investigate this using PCR methods?

The authors are comparing complete genetic knockout in animals with patient derived cells harbouring point mutations. Is it known what are the consequences of these point mutations? Do they affect protein levels/activity?

Are there any differences in the treatments in Figures 3G&J? Were the cells EGF-starved in 5G? If so, one would expect EGFR levels to be similar between control and patient cells as observed in 3J. This is important as increased EGFR degradation could reflect its higher activity. The authors should probe for activated EGFR (pEGFR) and compare it between control and patient cells. Downstream PI3K pathway should also be tested by probing for pAKT. In these experiments, cells should be EGF-starved prior to stimulation with EGF.

A more thorough investigation of EGFR recycling is required to explain its reduced levels/signalling in patient cells. It would be important to do EGFR localization with late endosome/lysosome and recycling endosome markers in order to investigate this.

Figure 3H: EGFR levels in this quantification should be normalized to T0 of the same cell line, rather than to control cells. This would give a better indication whether the degradation rates between control and patient cells are different. The colour coding of the graph lines should be clarified.

Can the authors look at EGFR levels or its downstream signalling in the WDR81^{-/-} mouse model for example by western blotting brain homogenates if antibodies are not available to detect EGFR by IHC?

Some methods could be better described in the figure legends/results section. For example, the fluorophore used to detect shRNA uptake in Figures 4A&B and how the neurodevelopmental defect was measured are unclear.

Figures 4A&B: could the difference in the neurodevelopmental defect be a result of a difference in shRNA targeting efficiency? The efficiency of shRNA target knockdown should be tested here.

Figure 4H: the authors conclude on page 6 that the defect in EGFR trafficking to early endosomes is particularly important at 120 min post EGF pulse. However, the authors show in Figure 3G that ERK signalling is reduced in patient cells as early as 5 min post stimulation. The authors should comment on this.

Figure 5: the authors should show whether the expression of cyclinD2 and its mutant can alter brain size of WDR81-mutant mice.

Minor comment:

Figure 2E: it would be informative to clarify what the arrows are pointing at in the IF images.

REVIEWER COMMENTS

Reviewer #1 (Remarks to the Author):

In this study, the authors provided data to suggest that Wdr81-mediated endosomal trafficking of EGFR plays a role in radial glial progenitor cell proliferation, and that Wdr81 deficiency leads to microcephaly. Although these findings provided additional evidence that WDR81 is important for neurogenesis, previous work has already shown that WDR81 is required for endosomal trafficking of EGFR and loss of WDR81 impaired neurogenesis in postnatal brain in mice. Thus, the current manuscript reports that WDR81 serves a somewhat similar purpose in radial glial cells as it does in HeLa cells and adult neural stem cells. Overall, the conceptual and mechanistic advance made by this study is limited.

We thank the reviewer for their review. The work was indeed based on previous studies on WDR81, but we make a series of important novel discoveries here, both in terms of mechanism and causes of the pathology. Moreover, we study here WDR81 in the relevant tissue and cells (radial glial cells in the developing neocortex) using a KO approach, as well as in patient cells, enabling to confirm that KO phenotypes are mirrored by disease-associated phenotypes. We demonstrate for the first time that WDR81 loss of function is associated with reduced proliferation, both in vivo and in patient fibroblasts. A central message is that reduced proliferation rates, in the absence of apoptotic cell death, can lead to primary microcephaly, which is highly unusual. We demonstrate that microcephaly-causing WDR81 loss of function can be rescued by a megalencephaly-causing mutation, well-known to drive excess proliferation. Therefore, the proliferation rate of radial glial progenitors is extremely sensitive and its imbalance can lead to severe cortical malformations.

We furthermore show that brain malformations associated with WDR81 mutation are due to its endosomal maturation role – and not its aggregophagy regulatory role – and that its mutation

affects the MAPK signaling pathway as a consequence of EGFR trafficking defects. We identify intracellular accumulation of EGFR and swollen early endosomes both in patient cells, and in vivo (new result: subcellular EGFR accumulation in radial glial cells, **Figures S3B-D**). We now demonstrate that WDR81 loss of function affects not only the degradation pathway, but also recycling to the plasma membrane (**Figure S3E and S3F**). Finally, we identify major neuronal positioning defects in WDR81KO brains, indicating that the endosomal role of WDR81 is required for neuronal migration, consistent with the lissencephaly observed in patients.

1. Microcephaly is a condition defined as a small head circumference present at birth (congenital or primary microcephaly) or later in life (postnatal or secondary microcephaly). Congenital microcephaly is usually caused by a decrease in the number of neurons generated during neurogenesis; and postnatal microcephaly is thought to result from the postnatal reduction of dendritic processes, myelination, and synaptic connections. In *Wdr81* mutant mice, the numbers of Pax6 apical progenitor and Tbr2 basal progenitor cells and subsequent neurogenesis are quite normal. While the author only showed that the *Wdr81* mutant mice displayed reduced brain size at P7, the etiology of microcephaly in *Wdr81* mutant need further analysis.

Patients carrying WDR81 mutations are born with microcephaly that progresses after birth. There is a graduation in the degree of primary microcephaly, the most severe identified in fetal cases with deceleration of head growth that led to medical pregnancy terminations (Cavallin et al, Brain, 2017). Therefore, WDR81 mutations cause both primary microcephaly and secondary microcephaly, which we now discuss in the manuscript. This study is focused on the embryonic roles of WDR81 in progenitor cells and, therefore, on the mechanism of primary microcephaly. To better analyze the etiology of the phenotype, we have now analyzed cortical thickness in WDR81 mutant mice at birth (P0), on top of P7. We show that, as in patients, P0 KO mice are microcephalic, although to a lesser extent than P7 mice (**Figure 1F**). To confirm this, we have now quantified the number of NEUN⁺ neurons at birth in mutant P0 pups, which reveals a ~ 30% reduction (**Figure 1G and 1H**). Therefore, as in patients, WDR81 KO mice display primary microcephaly that progresses after birth.

2. *Wdr81* mutation specifically alters proliferation of radial glial progenitors at mid and late, but not early, neurogenic stages. Is this a result of the distinct tempo-spatial expression of WDR81 during embryonic brain development?

We have now performed QPCR analysis of WDR81 expression throughout development, and show that WDR81 is indeed differentially expressed during development (**Figure S1A**). WDR81 expression is low at E12.5, when no proliferation defects are observed in mutants, and rise strongly from E14.5 onwards, when RG cell proliferation is altered. This data shows that WDR81 expression follows the responsiveness of RG cells of EGF, and indicates that alteration of RG proliferation in WDR81 mutants is a consequence of its temporal expression.

3. Cyclin D2 being localized to the endfoot of APs acts as a “basal fate determinant.” During asymmetric cell division, one of the daughter cells inherits its basal process, which automatically leads to asymmetrical inheritance of the cyclin D2 protein between the daughter cells. The daughter cell with cyclin D2 then becomes an AP, and the other without cyclin D2 becomes a neuronal cell or an intermediate progenitor (Tsunekawa et al 2012 EMBO J). In this study, the author showed that the Cyclin D2 overexpression rescued the progenitor proliferation in *Wdr81* mutant mice. However, is Cyclin D2 expression affected in *Wdr81* mutant radial glial cells? In addition, is the asymmetric division of radial glial cells affected in *Wdr81* mutant mice?

This is indeed a very interesting point. However, we want to point out that our model is not that Cyclin D2 acts downstream of EGFR-WDR81, but that proliferation imbalance, caused either by WDR81 or Cyclin D2 mutations, can be at the root of microcephaly and megalencephaly. We have unfortunately not managed to probe Cyclin D2 levels in the developing neocortex. We have tested several antibodies, but none of them displayed convincing specific signal in immunofluorescence or western blot. However, as suggested by reviewer 2, we have investigated in patient cells the levels of p-AKT, an upstream regulator of Cyclin D2 stability. This analysis revealed that, unlike p-ERK, p-AKT levels are not altered in WDR81 mutants (**Figure S2C**). These results suggest that WDR81 mutation may not alter Cyclin D2 stability. In support of this notion, we do not detect obvious defects in asymmetric divisions. As shown in **Figure 2G**, we have measured the relative amounts of RG cells, intermediate progenitors and neurons. Defects in asymmetric cell division is expected to lead either to a premature depletion of progenitors and a corresponding temporary increased neuronal production, or to increased maintenance of progenitors and the expense of neuronal production. We have not detected major fate abnormalities, except a small decrease in the percentage of neurons, that was however not mirrored by increase in progenitor numbers. In any case, these results are not consistent with premature neuronal differentiation as a cause of the microcephaly phenotype.

Reviewer #2 (Remarks to the Author):

Carpentieri et al present data investigating the role of WDR81 during microcephaly. They generate mice lacking WDR81 and investigate brain size and neuronal cell development, growth and survival. They show that mice lacking WDR81 exhibit a microcephaly phenotype and they attribute this effect to a prolonged S-phase of radial glial progenitors. The authors also investigate fibroblasts derived from patients and harbouring WDR81 mutations. They conclude from these experiments that in patient cells, EGFR levels, signalling and trafficking are altered when compared to control cells. Finally, the authors show that the expression of a degradation-resistant cyclinD mutant can reverse the mitotic defect observed in WDR81^{-/-} mice.

Overall, this is an interesting study. However, some of the findings are not investigated in depth and require further examination. More specific comments are as follows:

The authors show WDR81 RNA levels but have not confirmed the efficiency of WDR81 knockout as some cells may still express full length protein. If antibodies are not available to do so, can the authors investigate this using PCR methods?

We thank the reviewer for their review. We have analyzed the mRNA levels of the two WDR81 isoforms in WT and WDR81 mutant brains using QPCR, which revealed that both isoforms are strongly depleted, likely due to nonsense mRNA decay (**Figure 1B**).

The authors are comparing complete genetic knockout in animals with patient derived cells harbouring point mutations. Is it known what are the consequences of these point mutations? Do they affect protein levels/activity?

Both patient lines correspond to compound heterozygous mutations. The patient 1 line bears the following point mutations: 1882C-T and 3713C-G. The 1882 mutation is located in between the BEACH domain and the transmembrane domain, and leads to the replacement of a Gln-coding codon to a stop codon. This was previously shown to lead to nonsense mRNA decay (Cavallin et al, 2017, Brain). The 3713 mutation leads to Pro to Arg change and is located on the extracellular part on the transmembrane domain. The patient 2 line bears the following mutations: 1582C-T and a 4-nucleotide duplication (4036_4041). The 1582 mutation is located in the BEACH domain and leads to the replacement of a His to a Tyr on the protein. The deletion

is located in the cytosolic part of the transmembrane domain and leads to the generation of a premature stop codon. For both of these mutations, as well as for the other patients, a combination of a strong mutation (Stop codon) and a predicated weaker mutation (point mutation) is observed. Severe fetal cases with two strong mutations have also been reported. In all cases, parents carrying only one mutation do not show any pathological alteration. We have better described this in the manuscript.

Are there any differences in the treatments in Figures 3G&J? Were the cells EGF-starved in 3G? If so, one would expect EGFR levels to be similar between control and patient cells as observed in 3J. This is important as increased EGFR degradation could reflect its higher activity. The authors should probe for activated EGFR (pEGFR) and compare it between control and patient cells. Downstream PI3K pathway should also be tested by probing for pAKT. In these experiments, cells should be EGF-starved prior to stimulation with EGF.

The differences in EGFR levels in figures 3G and 3J are indeed a direct consequence of a different treatment of the cells. In 3G, cells were EGF-starved for 2H, as classically done to monitor EGFR downstream activation. Next, in figure 3J, we asked whether reduced EGFR levels were a consequence of its own activation or totally independent (for example due to global transcriptional effects in WDR81 mutant cells). To test this, we performed 24H EGF-starvation, in order to test the long-term effects of inactive EGFR on its own levels. This experiment indicated that EGFR levels were rescued, and therefore that EGFR level defects were a direct consequence of alterations in the EGFR pathway. We apologize if this was not clear enough, and have rephrased it in the manuscript. The reduced EGFR levels in cells at steady state make it difficult to interpret p-EGFR levels. We have however further probed the downstream PI3K pathway, which did not reveal any difference in p-AKT levels (**Figure S2C**). These results indicate that EGFR signaling defects affect the downstream MAPK pathway, but not the PI3K pathway.

A more thorough investigation of EGFR recycling is required to explain its reduced levels/signalling in patient cells. It would be important to do EGFR localization with late endosome/lysosome and recycling endosome markers in order to investigate this.

In order to probe intracellular processing of EGFR, we have used fluorescently-tagged EGF, which allows to follow its intracellular processing in a timely controlled manner. One caveat of this classical assay is that the high EGF doses used to visualize it trigger massive targeting of EGFR to degradation with very limited recycling. We however agree that investigating recycling in these cells is a crucial point. To do this, we have performed a transferrin recycling assay. Cells were treated with fluorescent transferrin, which is well-known to trigger endocytosis of the transferrin receptor and its massive recycling back to the plasma membrane. We show here that in WDR81 mutant fibroblasts transferrin is internalized but, as for the EGFR, is strongly delayed within EEA1-positive endosomes (**Figures S5E and S5F**). These results indicate that the endosomal maturation defects observed in WDR81 mutant cells affects both the degradation and recycling pathways, and suggest that both degradation and recycling of EGFR are affected.

Figure 3H: EGFR levels in this quantification should be normalized to T0 of the same cell line, rather than to control cells. This would give a better indication whether the degradation rates between control and patient cells are different. The colour coding of the graph lines should be clarified.

The purpose of the figure is to highlight the differential EGFR levels, as a cause of reduced MAPK signaling which is why we chose this representation, that directly reflects EGRF levels. We agree that this alternative normalization would emphasize the degradation kinetics,

however, given the extremely different levels of EGFR at T0, we find this difficult to compare. We therefore maintained the current representation, and addressed EGFR degradation kinetics in **Figure 4**.

Can the authors look at EGFR levels or its downstream signalling in the WDR81^{-/-} mouse model for example by western blotting brain homogenates if antibodies are not available to detect EGFR by IHC?

We have investigated EGFR levels in control vs WDR81^{-/-} brains. Western blotting of homogenates was somewhat inconclusive, with no reproducible variations of EGFR levels. A caveat of this experiment is that these extracts contain many different cell types, including neurons, which makes it difficult to probe EGFR levels in radial glial cells specifically. To overcome this, we have therefore performed immunostainings in brain tissue focusing on radial glial cells. We observe that intracellular EGFR foci are bigger in WDR81^{-/-} radial glial cells, but also more numerous, indicating an intracellular accumulation of EGFR (**Figures S3B, S3C and S3D**). These results indicate that intracellular processing of EGFR is similarly altered both in patient fibroblasts, and in mouse WDR81^{-/-} radial glial cells.

Some methods could be better described in the figure legends/results section. For example, the fluorophore used to detect shRNA uptake in Figures 4A&B and how the neurodevelopmental defect was measured are unclear.

We have now further detailed these experiments. In particular, we better indicate the plasmids that were electroporated, as well as how VZ+SVZ, IZ and CP were determined

Figures 4A&B: could the difference in the neurodevelopmental defect be a result of a difference in shRNA targeting efficiency? The efficiency of shRNA target knockdown should be tested here.

We now report shRNA KD efficiency at the mRNA level for WDR81 KD, WDR91 KD and p62 KD (**Figure S3A**). All three plasmids lead to efficient KD of their target, indicating that the phenotypic difference is indeed due to a more critical role of WDR81 endosomal maturation function during brain development.

Figure 4H: the authors conclude on page 6 that the defect in EGFR trafficking to early endosomes is particularly important at 120 min post EGF pulse. However, the authors show in Figure 3G that ERK signalling is reduced in patient cells as early as 5 min post stimulation. The authors should comment on this.

We apologize if this was confusing. The difference is once again due to the experimental setups, which were designed to ask different questions. In figure 3G, we tested whether the ERK signaling pathway was affected in mutant cells at steady state (in which EGFR levels are reduced). We performed a short EGF starvation, followed by a pulse of EGF to monitor the immediate activation of the pathway. In figure 4H, we wanted to test whether the reduced EGFR levels could be a consequence of endosomal processing defects. To do so, we wanted both WT and mutant cells to start with equal amounts of EGFR, and monitor EGFR intracellular behavior following EGF pulse. To do so, we EGF-starved the cells for 24 hours, to rescue EGFR levels (**see figures 3J and 3K**), and performed the assay. Intracellular processing of EGFR occurs on a different timescale than activation of the downstream signaling pathway (hours vs minutes). We have reworked the manuscript to explain this in a clearer manner.

Figure 5: the authors should show whether the expression of cyclinD2 and its mutant can alter brain size of WDR81-mutant mice.

This is a very interesting point. However, this would require to engineer a cyclin D2 knock-in mice carrying the mutations. Indeed, sparse in utero electroporation is insufficient to lead to alteration in brain size in mouse.

Minor comment:

Figure 2E: it would be informative to clarify what the arrows are pointing at in the IF images. The arrows point to cells double positive for EdU and BrdU. We have clarified this in the legend.

REVIEWERS' COMMENTS

Reviewer #2 (Remarks to the Author):

Altogether, the mechanistic data in this manuscript show that both EGFR and Transferrin accumulate at early endosomes in cells lacking WDR81. This demonstrates an apparent defect of the endocytic pathway in the absence of WDR81. Indeed, the difference in signalling between ERK and AKT could also support such conclusion. However, the main points raised in the initial review regarding EGFR degradation and recycling have not yet been addressed in order to explain its reduced levels. EGFR degradation in Fig. 3H was not properly analysed as requested and therefore one cannot conclude any defects in its degradation in the absence of WDR81.

In the newly added Figures S3E and S3F, the authors assess the co-localisation of transferrin with EEA1 which does not measure its recycling. Rather, the authors should have performed a chase experiment or localized transferrin with recycling endosomes. It is also important to keep in mind that defects in EGFR recycling do not always correlate with defects in transferrin recycling. Antibodies to detect EGFR in cells are widely available and biotinylation experiments can be used to look at surface expression.

Overall, the conclusions of the manuscript should be better aligned with the data presented. The manuscript could be simply modified to edit the text to ensure that the conclusions are supported by their findings.

Figure S3A: which RNA is the control sample measuring? This is unclear from this figure.

Reviewer #3 (Remarks to the Author):

I believe that the current version of the manuscript has been improved. The authors adequately addressed most of reviewer 1 concerns, providing new experimental data to support their claims. There is an interesting parallelism between the results obtained using KO mice and those derived from patients displaying mutations in WDR81. Overall, the current version of the manuscript convey an important message regarding the physiological roles of WDR81.

Reviewer #2 (Remarks to the Author):

Altogether, the mechanistic data in this manuscript show that both EGFR and Transferrin accumulate at early endosomes in cells lacking WDR81. This demonstrates an apparent defect of the endocytic pathway in the absence of WDR81. Indeed, the difference in signalling between ERK and AKT could also support such conclusion. However, the main points raised in the initial review regarding EGFR degradation and recycling have not yet been addressed in order to explain its reduced levels. EGFR degradation in Fig. 3H was not properly analysed as requested and therefore one cannot conclude any defects in its degradation in the absence of WDR81. In the newly added Figures S3E and S3F, the authors assess the co-localisation of transferrin with EEA1 which does not measure its recycling. Rather, the authors should have performed a chase experiment or localized transferrin with recycling endosomes. It is also important to keep in mind that defects in EGFR recycling do not always correlate with defects in transferrin recycling. Antibodies to detect EGFR in cells are widely available and biotinylation experiments can be used to look at surface expression.

Overall, the conclusions of the manuscript should be better aligned with the data presented. The manuscript could be simply modified to edit the text to ensure that the conclusions are supported by their findings.

We thank the reviewer for his re-evaluation of the manuscript. We agree with the reviewer's final point that the EGF⁵⁵⁵ and the transferrin⁵⁴⁶ assays may not unequivocally allow to claim EGFR recycling or degradation defects in WDR81 mutants. While none of these statements were present in the previous version of the manuscript, we have modified the text, in particular to indicate that we detected transferrin processing defects, rather than recycling defects (line 202).

Figure S3A: which RNA is the control sample measuring? This is unclear from this figure.

This representation was indeed confusing. We now show WDR81, WDR91 and p62 relative mRNA levels compared to a scramble control normalized to 1 (this normalized control corresponds to WDR81, WDR91 or p62 mRNA, depending on the shRNA tested).

Reviewer #3 (Remarks to the Author):

I believe that the current version of the manuscript has been improved. The authors adequately addressed most of reviewer 1 concerns, providing new experimental data to support their claims. There is an interesting parallelism between the results obtained using KO mice and those derived from patients displaying mutations in WDR81. Overall, the current version of the manuscript convey an important message regarding the physiological roles of WDR81.

We thank the reviewer for his positive evaluation of our manuscript.